# Comparing the Effects of Intracellular and Extracellular Magnetic Hyperthermia on the Viability of BxPC-3 Cells

**DOI:** 10.3390/nano10030593

**Published:** 2020-03-24

**Authors:** Gary Hannon, Anna Bogdanska, Yuri Volkov, Adriele Prina-Mello

**Affiliations:** 1Nanomedicine and Molecular Imaging Group, Trinity Translational Medicine Institute, Dublin 8, Ireland; hannonga@tcd.ie (G.H.); abogdans@tcd.ie (A.B.); 2Laboratory of Biological Characterization of Advanced Materials (LBCAM), Trinity Translational Medicine Institute, Trinity College Dublin, Dublin 8, Ireland; 3Advanced Materials and Bioengineering Research (AMBER) centre, CRANN institute, Trinity College Dublin, Dublin 2, Ireland; 4Department of Histology, Cytology and Embryology, First Moscow State Sechenov Medical University, Moscow 119992, Russia

**Keywords:** magnetic hyperthermia, iron oxide nanoparticles, magnetic nanoparticles, cancer

## Abstract

Magnetic hyperthermia involves the use of iron oxide nanoparticles to generate heat in tumours following stimulation with alternating magnetic fields. In recent times, this treatment has undergone numerous clinical trials in various solid malignancies and subsequently achieved clinical approval to treat glioblastoma and prostate cancer in 2011 and 2018, respectively. However, despite recent clinical advances, many questions remain with regard to the underlying mechanisms involved in this therapy. One such query is whether intracellular or extracellular nanoparticles are necessary for treatment efficacy. Herein, we compare the effects of intracellular and extracellular magnetic hyperthermia in BxPC-3 cells to determine the differences in efficacy between both. Extracellular magnetic hyperthermia at temperatures between 40–42.5 °C could induce significant levels of necrosis in these cells, whereas intracellular magnetic hyperthermia resulted in no change in viability. This led to a discussion on the overall relevance of intracellular nanoparticles to the efficacy of magnetic hyperthermia therapy.

## 1. Introduction

Iron oxide nanoparticles (IONP) have been researched extensively in a wide range of oncological applications [1,2]. One such promising field is magnetic hyperthermia [3]. This treatment involves the use of IONP to generate heat in tumours following stimulation with alternating magnetic fields (AMF). Under the influence of AMF, IONP rapidly change their polarity resulting in elevations of heat by way of hysteresis losses and relaxation losses [4,5]. At sufficiently small sizes (<50 nm), IONP consist of single magnetic domains and, in this case, heat is generated primarily through Nèel and Brownian relaxation [6,7,8]. Single domain IONP can also exhibit superparamagnetic properties characterized by thermally-driven inversions in their internal magnetisation [7,9]. Superparamagnetic IONP are ideal materials for magnetic hyperthermia as they have large magnetic moments that align uniformly to the applied AMF generating a substantial saturation magnetization, all while having a recorded average remnant magnetization of zero when the field is removed. This allows for a controlled thermal treatment that is favourable for in vivo applications [10,11]. 

Significant advances have been made in the last two decades with regards to the use of superparamagnetic IONP to treat cancer through magnetic hyperthermia. Since Gilchrist’s foundational work in this field in the 1950s and 60s [12,13], IONP design has improved to maximise biocompatibility while enhancing heating performance. Additionally, a wealth of in vivo efficacy studies and clinical trials have confirmed the huge potential of this treatment, which has led to the clinical approval of Nanotherm®, the first IONP to achieve regulatory approval for treating cancer with magnetic hyperthermia [5,14,15]. Despite these advances, many questions related to the underlying mechanisms of this therapy remain. One such question is the role that intracellular and extracellular IONP play in this therapy and whether one or both are necessary for treatment efficacy. Many papers have sought to functionalise IONP with targeting agents to improve uptake into cancer cells and enhance treatment efficacy; however, these efforts have generated mixed results [16,17]. Herein, we investigate the effects of intracellular and extracellular magnetic hyperthermia with BxPC-3 pancreatic adenocarcinoma cell line. Extracellular magnetic hyperthermia at temperatures of 40–42.5 °C showed effective induction of necrosis in these cells whereas intracellular magnetic hyperthermia demonstrated no therapeutic effect, leading to questions on its true relevance in the overall contribution to treatment efficacy.

## 2. Materials

Unless otherwise stated, all materials used were purchased from Sigma Aldrich, Wicklow, Ireland. The superparamagnetic IONP used in this study were fluidMAG/C11-D magnetite nanoparticles coated in a starch matrix and supplied by Chemicell, GmbH (Berlin, Germany) as part of the NoCanTher project (grant agreement no. 685795). These nanoparticles have been autoclaved by the suppliers to ensure sterility. The material is used as received.

## 3. Methods

### 3.1. Characterisation

The hydrodynamic diameter of the nanoparticles was assessed using nanoparticle tracking analysis (NS500 Nanosight, Malvern-Panalytical, Malvern, UK) according to protocols published previously [18,19], and now an established protocol for the EUNCL (EU Nanomedicine Characterisation Laboratory) [20]. These results were confirmed with dynamic light scattering (DLS) measurements (Malvern Nano- ZS, Malvern-Panalytical, Malvern, UK), following the EUNCL protocol for DLS size analysis [21]. Zeta potential data was provided by the nanoparticle supplier (at pH 7), Chemicell. For determining the dry diameter of the nanoparticles, transmission electron microscopy was used. Here, nanoparticles were diluted 1 in 1000 from the stock (100 mg/mL) in double distilled water (ddH_2_0) and adhered to Lacey carbon grids (AGAR Scientific, Stansted, UK). Images were taken using the JOEL 2100 (JOEL, Tokyo, Japan) at an acceleration of 200 kV and a beam current of 100–110 µA. The longest diameter of 200 individual nanoparticles was measured using ImageJ software to generate a size distribution (for a description of hydrodynamic size, zeta potential and dry size analysis, see Appendix A). 

### 3.2. Iron Quantification

Iron concentration was determined by atomic absorption spectroscopy (SpectraAA-200, Varian, California, CA, USA). Here the nanoparticles were dissolved in 1 mL 67–69% trace-element nitric acid (Fisher, Altrincham, UK) at a concentration of 125 µg/mL of nanoparticle and heated to 60 °C for four hours. The nanoparticles were then diluted in trace-elemental water (Fisher, Altrincham, UK) for analysis. A standard curve between 0–2.5 mg/L of iron was generated, and the concentration of the nanoparticles was determined from the average of three readings.

### 3.3. Heating Performance

The heating efficiency of these nanoparticles in response to an AMF was measured using a Five Celes inductor system (Five Celes, Lautenbach, France). This system uses a six-turn, moulded solenoid coil with an internal diameter of 71 mm. In this experiment, the nanoparticles were dispersed in 100 µL of double distilled water (ddH_2_O) at a nanoparticle concentration of 5 mg/mL (368 µg Fe/mL) and exposed to 35 mT, at a frequency of 92 kHz (parameters used for in vitro magnetic hyperthermia) for 60 s. Temperature changes were monitored using Optocon® fiber optic temperature sensors (Optocon®, Dresden, Germany) with an accuracy of ± 0.2 °C. From these temperature changes, specific absorbance rate (SAR) and intrinsic loss power (ILP) were calculated as reported in Kallumadil et al. [22].

### 3.4. In Vitro Cytotoxicity

BxPC-3 cells (ATCC, CRL-1687; Pancreatic adenocarcinoma of a female human aged 61) were cultured at 37 °C and 5% CO_2_ in RPMI 1640 media supplemented with 10% FBS and 1% penicillin-streptomycin (Invitrogen, Altrincham, UK) at 10,000 U/mL and 10,000 µg/mL respectively. Cells were seeded at 10,000 cells/well in a 96 well plate for 24 h. The cells were then washed with PBS and treated with either nanoparticles (12.5, 25, 50, 100, and 200 µg/mL; 200 µg/mL of nanoparticle corresponded to 147 µg Fe/mL), tacrine at 100 µM (positive control for organelle damage [23]), valinomycin at 120 µM (positive control for membrane damage [24,25]), or media alone (untreated) for 72 h. Following treatment, the cells were washed three times with PBS to remove extracellular nanoparticles and stained with 50 µL of Lysotracker® red (acidic organelle stain) and YO-PRO®-1 (membrane permeability stain) at 3.5 µL and 0.35 µL per ml of media respectively for 30 min at room temperature (stains supplied by Thermo scientific, Altrincham, UK and are described in Table 1). Lysotracker^®^ red stains acidic organelles, with increases in intensity corresponding to pH, and indicative of nanoparticle localisation into lysosomes. Decreases in intensity, however, is a marker for lysosomal damage [26,27]. YO-PRO®-1 is a green dye that measures cell membrane permeability; its localisation into the nucleus is used as a measure of cytotoxicity [27]. After staining for 30 min, the dyes are removed, and the cells are fixed with 100 µL of 3.7% formaldehyde for 20 min at room temperature. Following fixation, 0.5 µL of Hoechst 33342/mL in wash buffer is added to each well at 100 µL for 10 min. After two subsequent washes, the cells could be imaged. Cytell automated imaging system (GE Healthcare, Buckinghamshire, UK) was used to record automated images of seven fixed fields for each well. These images were analysed through high content screening analysis via InCell Investigator software (GE Healthcare, Buckinghamshire, UK) and intensity and morphology values for cell count, nuclear intensity, and organelle intensity were obtained and compared against untreated cells as measures of cytotoxicity.

### 3.5. Cell Uptake and Prussian Blue Staining

To quantify the levels of nanoparticle internalised into the cells, 1 × 10^5^ cells/well were seeded for 24 h in a 12 well plate. The cells were then treated with 200 µg/mL of nanoparticles for 24 h. After this time, the cells were washed with PBS, detached with trypLE and washed twice more with PBS and centrifugation steps (1000 rpm -at 94 RCF- for 5 min). Finally, the cells were counted using trypan blue staining and dissolved in 67–69%, trace-element nitric acid (Fisher, Altrincham, UK) at 60 °C for four hours for subsequent inductively coupled plasma–optical emission spectrometry (ICP-OES) analysis (Liberty 150, Varian, California, CA, USA). For this measurement, each treatment group is pooled within each experiment (five experiments in triplicate) to generate enough iron to be detectable by the instrument. The solution is then made up to 5 mL in trace-element water (Fisher, Altrincham, UK). ICP-OES was performed on these pooled samples as well as six iron standards in duplicate (from 0–2.5 mg/L). To complement this data, Prussian blue staining (iron- specific stain) was additionally performed to provide representative images of this iron uptake and identify where the nanoparticles are localised. The experiment was performed as before except, following 24 h treatment with IONP, the cells were washed three times with PBS, fixed with 4% formaldehyde for 20 min at room temperature, washed again as before, stained 1:1 with 4% HCl and 4% potassium ferrocyanide, washed again, and finally stained with Eosin for 3 min followed by an additional wash step. Bright field images were then taken using the Lionheart FX microscope (BIOTEK, Bad Friedrichshall, Germany) at 10× magnification.

### 3.6. In Vitro Magnetic Hyperthermia 

For this experiment, intracellular hyperthermia was compared against extracellular hyperthermia and a combination of intracellular and extracellular hyperthermia. Intracellular hyperthermia involves the exposure of intracellular IONP to AMF (IONP allowed to internalise for 24 h) while extracellular hyperthermia only exposes extracellular IONP (IONP added to the cell media directly before) to AMF. Intracellular and extracellular hyperthermia contains both. A summary of this experiment is demonstrated in Figure 1 below. BxPC-3 cells were seeded at 1 × 10^5^ cells/well in a 12 well plate. After 24 h, cells to be treated with intracellular nanoparticles alone, intracellular hyperthermia, intracellular and extracellular nanoparticles alone, or intracellular and extracellular hyperthermia were treated with 200 μg/mL of nanoparticles. At 48 h, cells are treated with magnetic hyperthermia. Here, the cells are washed with PBS, detached with trypLE and transferred into 1.5 mL Eppendorf tubes. They are then washed again in PBS through a centrifugation step and redispersed in 100 µL of either media (for intracellular hyperthermia or magnetic field alone), or 500 µg of nanoparticle in 100 µL media (for extracellular nanoparticles alone, extracellular hyperthermia, intracellular and extracellular nanoparticles alone, or intracellular and extracellular hyperthermia). For the cells exposed to 500 µg of nanoparticle alone, they are kept at 37 °C for 30 min. For the cells exposed to the magnetic field, the tubes are wrapped in parafilm and a sterile fibre optic temperature probe (washed in 70% ethanol) is pierced through the film into the cells and media. A water bath keeps the cells at 37 ± 1 °C before the AMF is applied. The cells are then exposed to a field of 35 mT at a frequency of 92 kHz. This field and frequency enabled the extracellular nanoparticles to reach therapeutic temperatures while also having no effect on cells alone (see results). Once the temperature in the media of the cells treated with extracellular nanoparticles reaches 40 °C (within 60–120 s), the timer is started and the cells are exposed to temperatures between 40–42.5 °C or ‘mild hyperthermia’ [28,29] for 30 min. Following AMF exposure, the cells are washed three times with their original RPMI media (centrifuged at 1000 rpm for 5 min) and placed back into a 12 well plate. After 24 ± 1 h, the media in each well was added to 1.5 mL tubes and cells are detached with TrypLE and added to their respective media to be stained for flow cytometry analysis (see Section 3.5). The cells are subsequently analysed by flow cytometry to identify populations or apoptotic and necrotic cells. As reported in Blanco-Andujar et al. [30] and Ludwig et al [31], the effects of magnetic hyperthermia on viability are the most reflective in the first 24 h, with changes in viability between 24 and 48 h proving to be negligible.

### 3.7. Apoptosis/Necrosis Detection

To detect levels of apoptosis and necrosis in the BxPC-3 cells after treatment, APC-Annexin V and 7-AAD stains (BioLegend, London, UK) were used as the nanoparticles were shown not to interfere in these channels (Appendix A). 24 h after the exposure to the AMF, the media and detached cells were washed twice with staining buffer and stained with 50 µL of APC-Annexin V (2.5 µL/mL of Annexin binding buffer) for 30 min. After such staining, the cells were washed three times with Annexin binding buffer and redispersed in 2.5 µL/mL of 7-AAD in staining buffer. The cells were then analysed with the FACSCanto II flow cytometer from BD Biosciences, San Jose, CA, USA (10,000 recorded events per treatment) and the data was subsequently analysed via FlowJo 10 and GraphPad Prism 7 software (gating strategy: Appendix A). Positive stain controls for APC-Annexin V were cisplatin treated cells at 50 µM for 24 h and for 7-AAD were 10% DMSO treated cells for 1 h. 

### 3.8. Caspase 3 Activity

Following apoptosis/necrosis detection by flow cytometry, caspase 3 activity was measured colourimetrically at 405 nm according to manufacturer’s protocol (Abcam ab39401, Cambridge, UK) to distinguish whether the main mechanism of cell death was apoptosis or necrosis. For this analysis, in vitro magnetic hyperthermia was repeated as before in four more experiments. In order to get enough protein for the analysis, each treatment group from two experiments had to be pooled together. For each experiment, BxPC-3 cells were also treated with 50 µM of cisplatin for 24 h as a positive control. 1 µg/µL of protein from each treatment group was tested in the assay and results were corrected for total protein concentration (BCA kit, Thermo Scientific, Altrincham, UK). Caspase 3 activity was presented as levels of absorbance at 405 nm in each treatment group against the untreated.

### 3.9. Statistical Analysis

All statistical analysis was done using GraphPad Prism 7 software. All results are reported as mean ± standard deviation. High content screening data was analysed via one-way ANOVA followed by Dunnett’s test. Apoptosis/Necrosis data was analysed by two-way ANOVA followed by Tukey’s test for multiple comparisons. Caspase activation was measured with one-way ANOVA followed by Dunnett’s test. Significance was represented by * *p* < 0.05; ** *p* < 0.01; *** *p* < 0.001; **** *p* < 0.0001. 

## 4. Results

### 4.1. Nanoparticle Characterization and Heating Performance

NTA reported a mean size of 100 ± 2.6 nm, which was confirmed with the closely resembled DLS measurements of 91.2 nm (PDI = 0.145). DLS measured an average zeta potential of −21 ± 5.86 mV at pH 7 (negative charge is due to phosphate groups bound to sugar moieties on the starch coating), while TEM images were analysed on ImageJ to determine a mean dry diameter of 11 ± 3 nm (based on 200 individual measurements of the largest diameter of each nanoparticle; see Appendix A). Upon exposure to 35 mT at 92 kHz for 60 s, the nanoparticles displayed a SAR of 98 W/g_Fe_ in water, corresponding to an ILP of 1.4 nHm^2^/kg (Table 2 and Figure 2). This heating capacity resembles other IONP used for in vitro magnetic hyperthermia in the literature [32,33,34]. Finally, the nanoparticles contained an average of 0.736 ± 0.01 mg of iron for every mg of total nanoparticle. 

### 4.2. In Vitro Cytotoxicity

Results acquired from the multiparametric analysis of the high content screening experiments showed no significant changes to the cell count, nuclear membrane permeability, or lysosomal permeability parameters after 72 h treatment of up to 200 µg/mL of nanoparticles against the untreated negative control, while positive controls (tacrine and valinomycin) quantitatively and visibly reduced cell count and lysosomal mass, while increasing nuclear permeability (Figure 3 and Figure 4). 

### 4.3. Cell Uptake and Prussian Blue Staining

Cellular uptake of the nanoparticles was assessed through qualitative and quantitative measures. ICP-OES identified an average uptake of 12.8 ± 3.6 pg Fe/cell (12.8 pg Fe corresponds to 16,487 nanoparticles, as measured by NTA) after 24 h. Prussian blue staining confirmed this internalisation and an accumulation of the nanoparticles around the nuclear membrane was consistently observed (Figure 5). 

### 4.4. Temperature Generation during In Vitro Magnetic Hyperthermia

Temperature graphs below show successful heating of the nanoparticles when used for extracellular hyperthermia (500 µg IONP in 100 µL media) and intracellular and extracellular hyperthermia with temperatures of 41.2 ± 0.6 °C being achieved across the entire experiment. No changes in media temperature were seen with intracellular nanoparticles in comparison to cells treated with the magnetic field and media alone. It is worth noting that the cells exposed to AMF alone never reached temperatures above 38 °C (Figure 6).

### 4.5. Cell Viability Detection Following Magnetic Hyperthermia

Following the hyperthermia treatments described previously, the cells were assessed for their viability by flow cytometry after 24 ± 1 h. Intracellular magnetic hyperthermia displayed no significant effect on the viability of the BxPC-3 cells when compared against the untreated, whereas extracellular magnetic hyperthermia—both alone and in combination with intracellular magnetic hyperthermia—showed significant reductions in cell viability after 30 min treatment. Additionally, the presence of the magnetic field alone had no effect on the cells either. This significant reduction in viability with extracellular magnetic hyperthermia is also observed when compared against the nanoparticles alone; therefore, the nanoparticles by themselves are not inducing this effect, but the whole magnetic hyperthermia treatment (Figure 7). Moreover, heating the BxPC-3 cells with an incubator to 42.5 °C for 30 min showed no significant effect on viability against untreated cells, confirming that the magnetic hyperthermia treatment as a whole was affecting the cells, and not just the inherent temperature elevation (Appendix A).

### 4.6. Caspase Activity

Differences in caspase activity were negligible in all treatment groups except for the positive control (Figure 8). Pairing this with the flow cytometry data suggests that the primary mechanism of cell death in this case was necrosis, as the cells stained positive for both Annexin and 7-AAD following extracellular magnetic hyperthermia treatment.

## 5. Discussion

FluidMAG/C11-D nanoparticles were characterized and assessed for their heating capabilities in response to an AMF of 35 mT at 92 kHz. The nanoparticles were deemed to have no effect on viability of BxPC-3 cells at concentrations up to 200 µg/mL in vitro using multiparametric analysis and so this concentration was used to evaluate their uptake into the cells. ICP-OES demonstrated that an average of 12.8 ± 3.6 pg Fe was internalised into each cell (with 12.8 pg Fe equating to 16,487 nanoparticles) and subsequent Prussian blue staining confirmed this internalisation and identified the nanoparticles propensity to accumulate around the nuclear membrane. This level of iron uptake into cells is comparable to the results reported from similar protocols published previously [35,36,37]. Next, intracellular hyperthermia, extracellular hyperthermia, and intracellular and extracellular hyperthermia were compared to define differences in viability through Annexin V/7-AAD staining. Intracellular hyperthermia showed no change in viability against the untreated, whereas cells exposed to extracellular nanoparticles and magnetic fields underwent significant apoptosis/necrosis (staining positively for both Annexin V and 7-AAD) against untreated cells and nanoparticle treated cells alone. This result was similar to that reported in Ludwig et al for BxPC-3 cells [31].

Cells that stain positively for 7-AAD have a permeabilised cellular membrane characteristic of cells that have undergone late apoptosis or necrosis, which allows 7-AAD to become internalised and intercalate to guanine and cytosine regions of DNA [38]. By contrast, Annexin V relies on the extracellular exposure of phosphatidyl serine (PS) from the plasma membrane—which normally faces internally in healthy cells—where it binds to PS in a calcium-dependant manner, acting as a positive early stain for apoptosis [39]. These two mechanisms of cell death can overlap if the cells undergoing apoptosis do not get phagocytosed, and so enter a stage of secondary necrosis which shares many features of primary necrosis [40,41]. In order to distinguish the primary mechanism of cell death following hyperthermia, caspase-3 activity was evaluated using colourimetric assay. Caspase-3 activity is essential for efficient apoptosis and so its expression levels against the untreated will indicate if apoptosis is occurring or not [42]. No changes in caspase-3 activity were identified following magnetic hyperthermia and so necrosis was deemed the primary mechanism of cell death in this case. This strong population of necrotic cells was also observed in the above-mentioned Ludwig et al. [31].

As intracellular magnetic hyperthermia resulted in no change in viability, this leads to questions over its relevance to this treatment as a whole. IONP internalised into cells experience an inhibition of Brownian motion and therefore heating capability [17,43]. Therefore, the efficacy observed with this treatment in vivo and the clinic may be solely due to extracellular nanoparticles as they are the primary source for temperature elevations in the tumour. Importantly, this has implications for IONP design and whether efforts to improve the internalisation of these nanoparticles through functionalisation with targeting agents are necessary. Nonetheless, further studies are needed to elucidate whether intracellular nanoparticles can contribute to treatment efficacy in alternative ways such as instigating anti-tumour immune responses or disrupting intracellular signalling pathways that may enhance the effects of combination therapies, as magnetic hyperthermia has been reported to do [44,45]. 

## 6. Conclusions

The effect of intracellular and extracellular magnetic hyperthermia on the viability of BxPC-3 cells was compared after a 30 min treatment. It was found that extracellular magnetic hyperthermia (at temperatures of 41.2 ± 0.6 °C) could induce significant levels of necrosis in these cells whereas intracellular magnetic hyperthermia showed no effect on viability. This therefore led to questions on the overall influence of intracellular IONP to treatment efficacy and whether they are necessary to achieve desired therapeutic effects. It is not yet known whether intracellular IONP plays a role in alternative anti-tumour effects such as stimulating anti-tumour immune responses or inducing alterations to intracellular signalling reported to be involved in magnetic hyperthermia treatment efficacy; therefore, future work is required to clarify this role. This research may provide indications for IONP design for magnetic hyperthermia applications in the future. 

## Figures and Tables

**Figure 1 nanomaterials-10-00593-f001:**
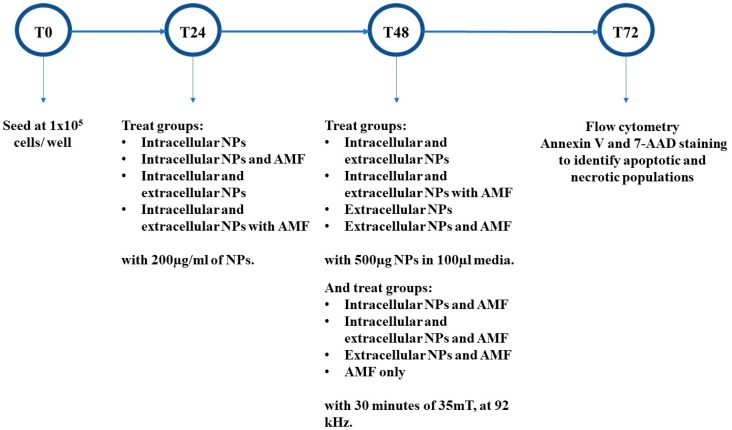
Summary of in vitro magnetic hyperthermia protocol. BxPC-3 cells are seeded at time zero (T0). Cells to contain intracellular nanoparticles are treated with 200 µg/mL at 24 h (T24). At 48 h (T48), cells exposed to extracellular nanoparticles are treated with 500 µg of nanoparticles in 100 µL media, and cells to be treated with magnetic fields, are exposed to 35 mT at 92 kHz for 30 min. Finally, at 72 h (T72), all treatment groups are stained with annexin V and 7-AAD and analysed by flow cytometry for detection of apoptotic and necrotic cells. Abbreviations: NPs, nanoparticles; AMF, alternating magnetic field.

**Figure 2 nanomaterials-10-00593-f002:**
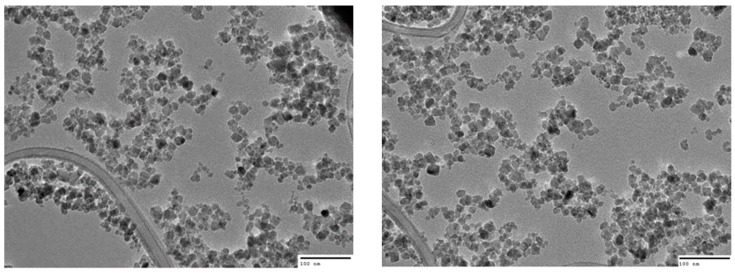
Representative TEM images of fluidMAG/C11-D nanoparticles. Captured with the JOEL 2100 (JOEL, Tokyo, Japan) at an acceleration of 200 kV and a beam current of 100−110 µA. Scale bar: 100 nm.

**Figure 3 nanomaterials-10-00593-f003:**
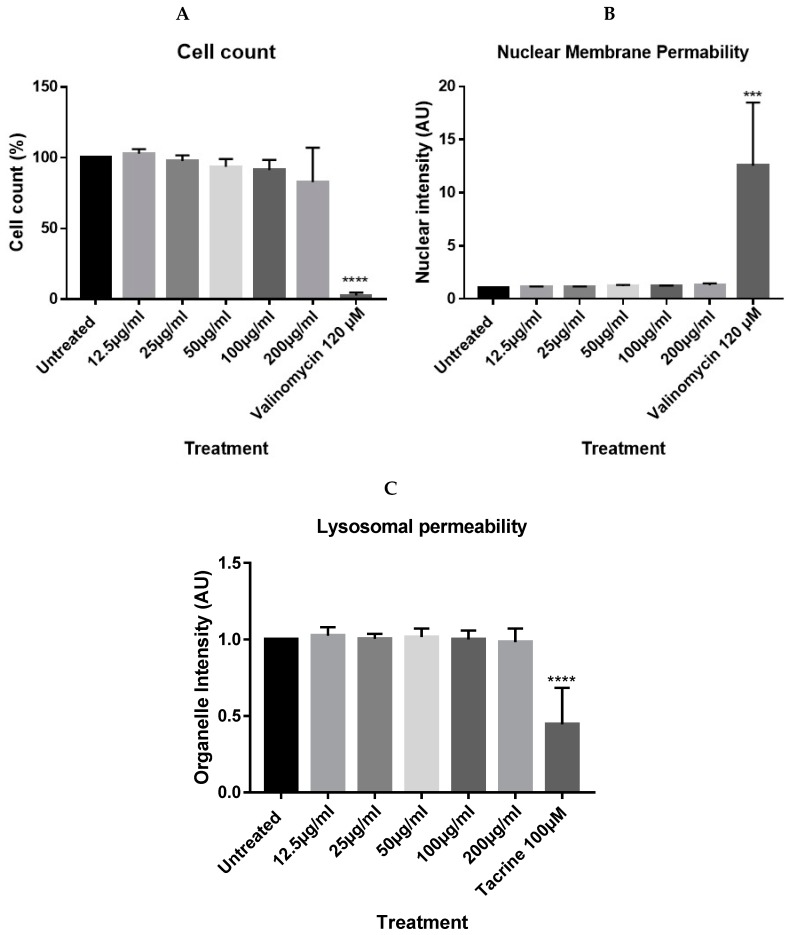
Cell count (**A**), nuclear (**B**), and organelle intensity (**C**) plots following high content screening analysis of BxPC-3 cells treated with fluidMAG nanoparticles. Graphs displaying differences in cell count, nuclear intensity, or organelle intensity after treatment with the nanoparticles up to 200 µg/mL, tacrine at 100 µM, and valinomycin at 120 µM. Cell count values are represented as a percentage of the untreated cells. Membrane permeability and organelle permeability values are normalised against the untreated cells. Results derived from three experiments in triplicate. Significance tested via one-way ANOVA followed by Dunnett’s test. Error bars = standard deviation. **** p < 0.0001, *** p < 0.001.

**Figure 4 nanomaterials-10-00593-f004:**
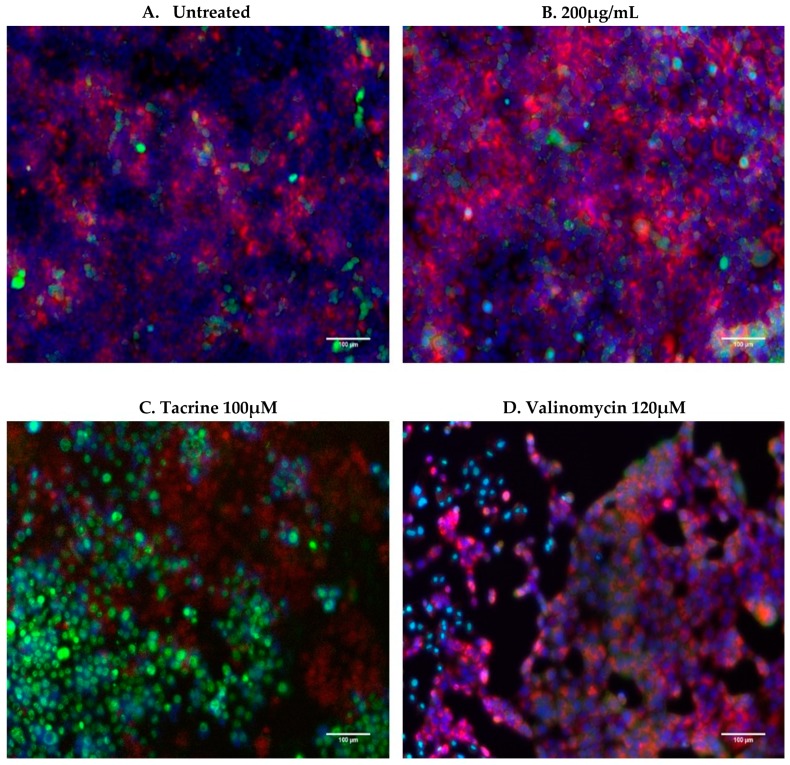
Representative merged images of BxPC-3 cells following treatment. Merged images of untreated (**A**), 200 µg/mL treated (**B**), and treated with tacrine (**C**) and valinomycin (**D**) positive controls. Scale bar: 100 µM. Both untreated and 200 µg/mL treated images depict high cell numbers and little green dye located in the nucleus of cells while background levels of red dye consistent between each group. Tacrine and valinomycin-treated cells on the other hand have a visibly lower cell count and both have a high presence of green dye in the nucleus. Additionally, tacrine treated cells have a lower incidence of red dye indicating lysosomal damage to these cells, as previously reported in [23].

**Figure 5 nanomaterials-10-00593-f005:**
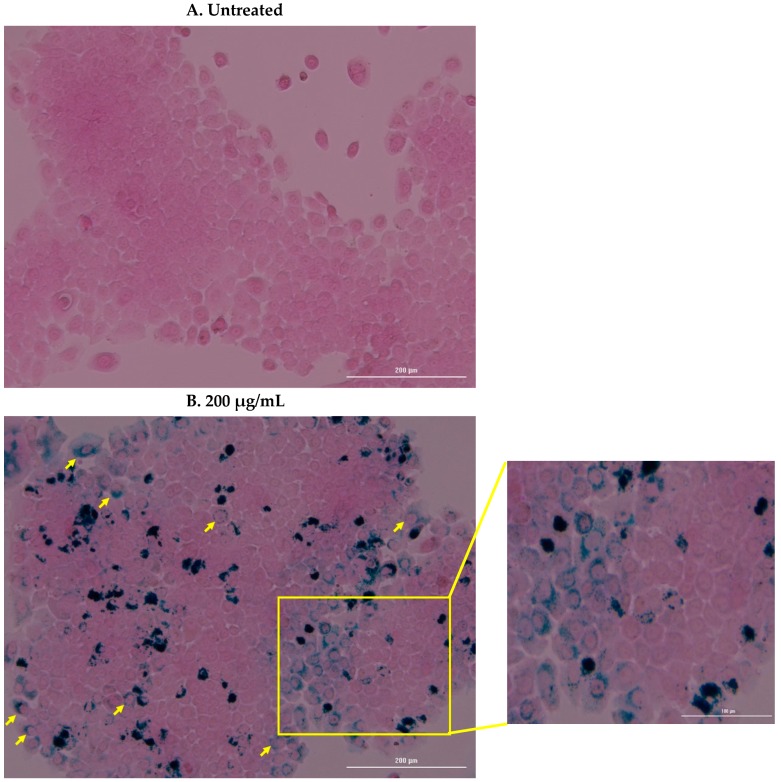
Prussian blue staining of untreated and treated BxPC-3 cells. Untreated cells showed negligible background iron, noted by the absence of blue staining in image (**A**). The blue staining in image (**B**) show the nanoparticles tendency to accumulate around the nucleus of the BxPC-3 cells, as highlighted by the yellow arrows and zoomed in image (**B**). This intracellular accumulation complements the results from the ICP-OES analysis. Scale bar: 200 µm for image (**A**) and (**B**), and 100 µm for zoomed in image (**B**).

**Figure 6 nanomaterials-10-00593-f006:**
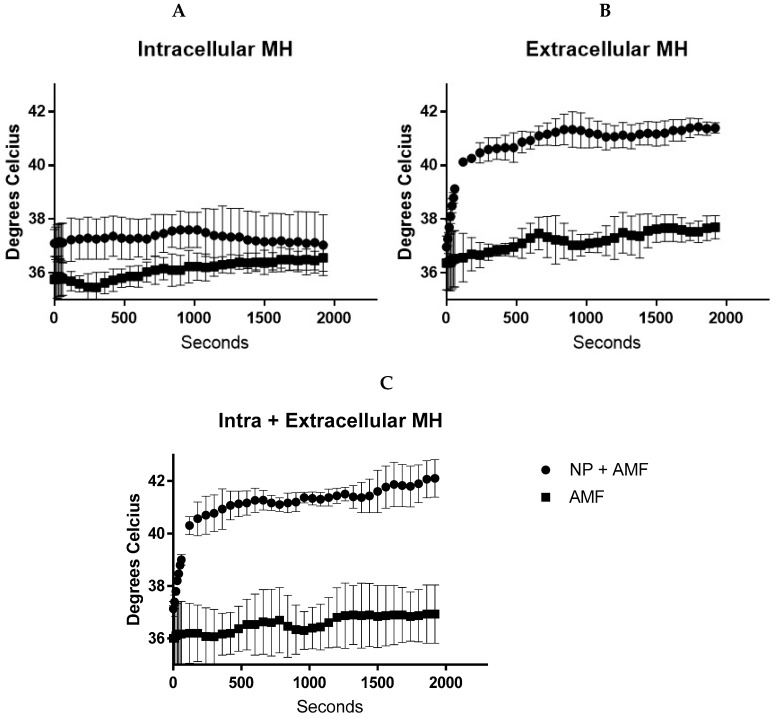
Temperature graphs of magnetic hyperthermia with BxPC-3 cells. Intracellular and magnetic field only cells remained at biologically viable temperatures (**A**). Extracellular nanoparticles successfully generated temperatures required for mild hyperthermia (**B**) and (**C**). Abbreviations: NP, nanoparticles; MH, magnetic hyperthermia; AMF, alternating magnetic field.

**Figure 7 nanomaterials-10-00593-f007:**
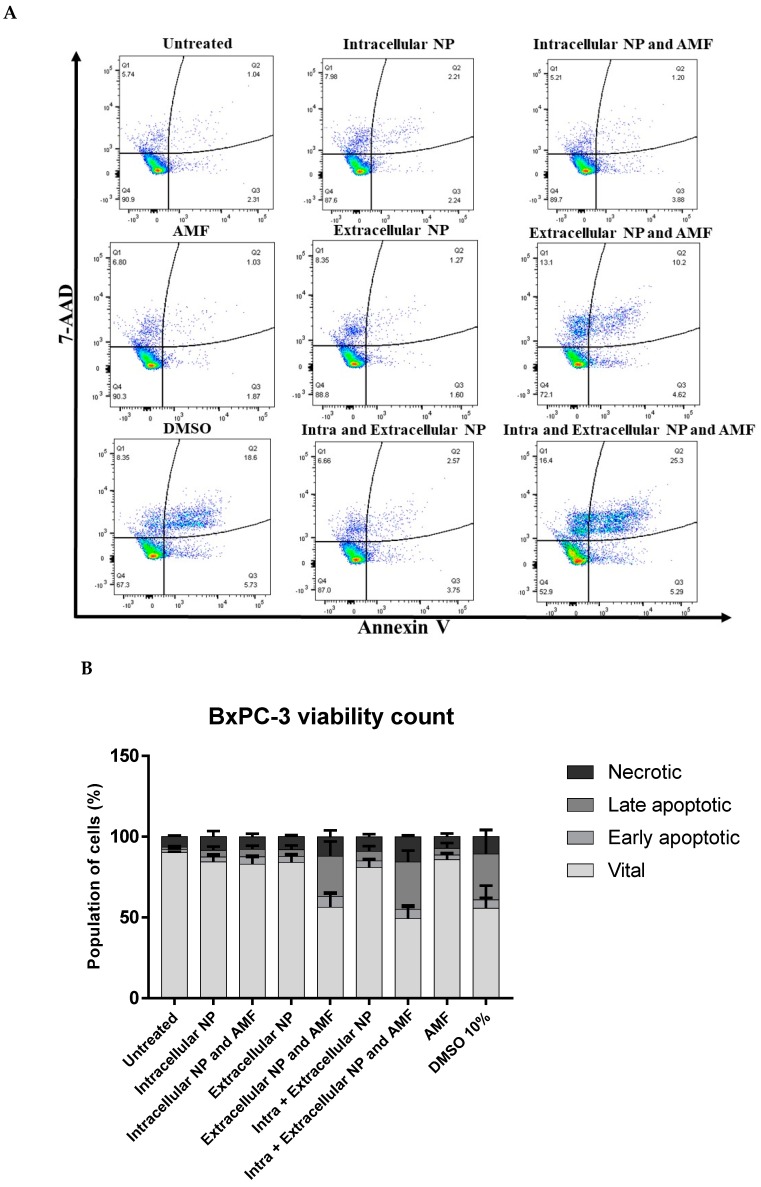
Viability of BxPC-3 cells after in vitro magnetic hyperthermia. (**A**) Representative flow experiment depicting Annexin V^+^/7-AAD^−^ (early apoptotic), Annexin V^+^/7-AAD^+^ (late apoptotic), 7-AAD^+^ (necrotic), and Annexin V^−^/7-AAD^−^ (vital) cells in each treatment group. (**B**) Graph comparing populations of vital, early apoptotic, late apoptotic, and necrotic cells in each treatment group. Results are from six individual experiments with 10,000 events recorded for each. Error bars = standard deviation. Abbreviations: NP, nanoparticle; Intra, intracellular; Extra, extracellular; AMF, alternating magnetic field. (**C**) Table describing significant differences in vital, early apoptotic, late apoptotic and necrotic cells from each treatment group against untreated cells. (**D**–**F**) Tables describing significant differences of these same cellular populations with nanoparticles exposed to AMF versus nanoparticles alone. Results are from six individual experiments with 10,000 events recorded for each. Abbreviations: NS, not significant; NP, nanoparticle; Intra, intracellular; Extra, extracellular; AMF, alternating magnetic field. Analysed using two-way ANOVA followed by Tukey’s test for multiple comparisons. **p* < 0.05. **** *p* < 0.0001.

**Figure 8 nanomaterials-10-00593-f008:**
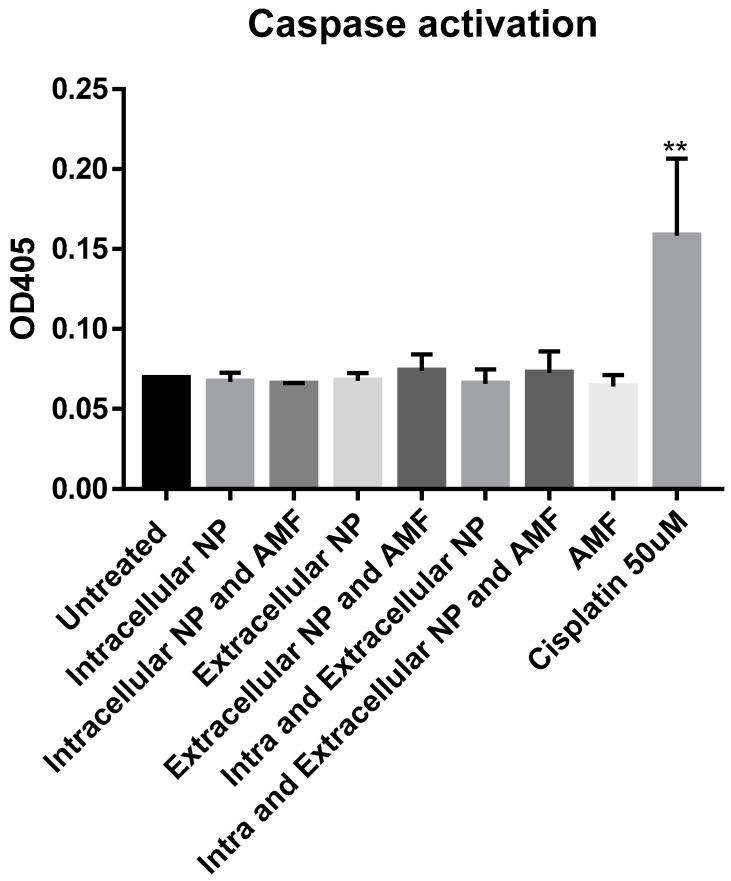
Caspase activation with BxPC-3 cells following magnetic hyperthermia treatment. Results are from four individual experiments, of which two experiments are pooled together to get the desired concentration of protein required for the assay. Values are a result of absorptions at 405 nm. Significance is against the untreated. Significance was assessed using one-way ANOVA followed by Dunnett’s test. Abbreviations: NP, nanoparticle; Intra, intracellular; Extra, extracellular; AMF, alternating magnetic field. Error bars = standard deviation. ** *p* < 0.01.

**Table 1 nanomaterials-10-00593-t001:** Description of each stain used for high content screening analysis

Stain	Excitation/Emission (nm)	Stock Concentration
Lysotracker® red	577/590	1 mM
YO-PRO®-1	491/509	1 mM
Hoechst 33342	350/461	16.2 mM

**Table 2 nanomaterials-10-00593-t002:** Summary of fluidMAG/C11-D characterization. Summary of characterization data by NTA, DLS, AAS, TEM, and heating capability assessment through SAR and ILP values. Zeta potential was provided by the supplier: Chemicell, GmbH. Values represented as mean ± standard deviation. Abbreviations: NP, nanoparticle.

Measured Parameter	Value (Technique)
Mean hydrodynamic size	100.0 ± 2.6 nm (NTA) 91.2 nm (DLS)
Nanoparticle number	9.48 × 10^13^ ± 4.60 × 10^12^ NP/mL (NTA)
Polydispersity index	0.145 (DLS)
Zeta potential	−21.0 ± 5.86 mV (DLS)
Mean dry size	11 ± 3 nm (TEM)
Specific absorbance rate	98 W/g_Fe_
Intrinsic loss power	1.4 nHm^2^/kg
Fe content	0.736 ± 0.01 mg Fe/mg NP (AAS)

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
