# Peer review of "Comparing the Effects of Intracellular and Extracellular Magnetic Hyperthermia on the Viability of BxPC-3 Cells"

_nanomaterials, 2020, doi:10.3390/nano10030593_

Round 1

Reviewer 1 Report

Authors should provide information on the following points:

1- In the introduction, they should rewrite the paragraph referring to superparamagnetism state (lines 31 to 34):

1.1-Discuss in terms of relaxation time instead of “to flips in its magnetic moment”

1.2 Avoid “magnetic moment of zero”. The magnetic moment is never zero in a magnetic material. Only diamagnetic materials has a atomic magnetic moment zero.

What is zero is the average of the remanent magnetization (not magnetic moment) measured with a technique whose data acquisition time (tm)  is greater than the relaxation time (tr) of superparamagntic nanoparticles (in this case no remanence and no coercivity are measured). In fact, hyperthermia is based on applying an alternating field of such a high frequency that the tm

It can be understood that for an interdisciplinary topic like the one presented in this paper, the authors are not experts in magnetism but errors in the basic and fundamental concepts of magnetism at the nanoscale have to be corrected.

2- The authors would have to justify the value of the magnetic field used and its frequency in the hyperthermia experiments. Have they tried other values?

3- The conclusions are too brief. Can the authors make an effort to expand on them?

Author Response

Dear Reviewer,

Thank you for reviewing our manuscript and providing comments.

We have addressed each of these in the file attached and hope it is to your satisfaction.

Yours sincerely,

The Authors.

Reviewer 2 Report

Reviewer:

Nanomaterials

Manuscript ID: nanomaterials-726781

Title: Comparing the effects of intracellular and extracellular magnetic
hyperthermia on the viability of BxPC-3 cells
Authors: Gary Hannon, Anna Bogdanska, Yuri Volkov, Adriele Prina-Mello *
Submitted to section: Biology and Medicines

Dear Editor:

The manuscript focused on the “Comparing the effects of intracellular and extracellular magnetic hyperthermia on the viability of BxPC-3 cells”, which is new novel and very useful in sensor fields. It is recommended to accept after major revision. However, some parts need to revise, which are listed below as follows. The main points need to revise before publication.

[1] The new relate references are needed to add in the revised manuscript.

[2] The authors investigate many parameters in this study. What is optimal condition in this work? Please explain and add it in the revised manuscript.

[3] The grammar of English should be written more carefully in the manuscript; English must be checked and improved by Native English speaker.

[4] What are the important applications in this study? Please add in the revised manuscript.

[5] The author should compare the specific properties of other materials and explain the benefits of this material.

[6] Please evaluate the feasibility of practical clinical applications

[7] Why iron-based cell can fight this disease? Is there any mechanism

? Please explain in details.

[8] In lines 279 to 280, tacrine treated cells have a lower incidence of red dye indicating lysosomal damage to these cells. Please explain the reason in details.

Sincerely yours.

Author Response

(The authors gave the same response as above.)

Reviewer 3 Report

“Comparing the effects of intracellular and extracellular magnetic hyperthermia on the viability of BxPC-3 cells” Gary Hannon, Anna Bogdanska, Yuri Volkov and Adriele Prina-Mello

The article goal is to compare the effects of intracellular and extracellular magnetic hyperthermia in BxPC-3 cells to determine the differences in efficacy between both. Magnetic hyperthermia involves usage of iron oxide nanoparticles to generate heat in tumours following stimulation with alternating magnetic fields.

The authors show that extracellular magnetic hyperthermia at temperatures between 40-42.5 °C could induce significant level of necrosis in these cells, whereas intracellular magnetic hyperthermia does not change cell viability.

The obtained data can make significant impact for applications of nanomaterials for  intracellular and extracellular magnetic hyperthermia.

The several remarks are below.

The IONP used in this study were fluid MAG/C11-D magnetite nanoparticles coated in a starch matrix. Questions: 1. What type of matrix was used? 2. Is this matrix thermal stable? The authors mentioned that the NP sterilization was ensured through autoclaving. The main thing: this particles were manufactured by the authors or were received  as complete material? More information should be in the part of Characterization of the hydrodynamic diameter, measuring and  dynamic light scattering (DLS): what types solution, pH, number of cycles etc.

Line 85 3.4. In vitro cytotoxicity paragraph needs to be verified and rewrite.

For example: lines 112-113 “The cells were then treated with 200 μg/ml of nanoparticles for 24 hours,   before being washed with PBS, detached with trypLE, and washed a further two times with PBS” Washed after detached by Centrifugation?

Some information included twice: line 200 µg/ml of nanoparticles (corresponding to 147 μg 138 Fe/ml) This sentence was repeated at list twice on the same page.

139 “Here, the cells are washed with PBS and detached with trypLE into 1.5 ml Eppendorf tubes (were the cells transferred after the dethaching?).

152 “After 24±1 hours, the cells are analyzed by flow cytometry to identify populations or apoptotic and necrotic cells (the cells were analyzed by flow cytometry into the wells or the cells were detached and collected first?)

-21.0± 5.86 mV (DLS) Why negative charge? Explain it please. Probably it is matrix charge.

Lines 137 – 143: “extracellular hyperthermia were treated with 200 µg/ml of nanoparticles (corresponding to 147 μg  Fe/ml)……or 500  μg of nanoparticle (368 μg Fe)  Questions: Why the concentration of NP are so different?

What physiological sense for these concentrations of iron, it is about 10 times higher than Iron concentration in erythrocytes (0.5 mg/mL).

In the manuscript, some pictures are not easy to read and understand, and they need to be verified also.  

For example,  Figure 3. “Cell count, nuclear and organelle intensity plots following high content screening analysis”.

Question: Nuclear membrane permeability and organelle permeability values are normalized against the untreated cells. However, number of cells treated by Valinomycin is very low, but “Nuclear intensity” for this cells is highest. How it was normalized?  

Conclusion should be rewriting, because hyperthermia (at temperatures of 41.2±0.6 °C) induced significant levels of necrosis in these cells. At this high temperature necrosis is obvious, this fact is not new.

Important result that intracellular magnetic hyperthermia did not affect viability.  This fact should be discussed in body text more profound.  

Major review of text  is required

Author Response

(The authors gave the same response as above.)

Round 2

Reviewer 2 Report

Dear Editor:

According to the revised version, it can be accepted and published in "Nanomaterials" journal.

Sincerely yours.

Reviewer 3 Report

Article in the all parts was significant improved, and new version of conclusion looks well.

My decision - article can be published in the present form.